# Bitter taste receptors confer diverse functions to neurons

Rebecca Delventhal, John R Carlson*

Department of Molecular, Cellular, and Developmental Biology, Yale University, New Haven, United States

**Abstract** Bitter compounds elicit an aversive response. In *Drosophila,* bitter-sensitive taste neurons coexpress many members of the Gr family of taste receptors. However, the molecular logic of bitter signaling is unknown. We used an in vivo expression approach to analyze the logic of bitter taste signaling. Ectopic or overexpression of bitter Grs increased endogenous responses or conferred novel responses. Surprisingly, expression of Grs also suppressed many endogenous bitter responses. Conversely, deletion of an endogenous *Gr* led to novel responses. Expression of individual Grs conferred strikingly different effects in different neurons. The results support a model in which bitter Grs interact, exhibiting competition, inhibition, or activation. The results have broad implications for the problem of how taste systems evolve to detect new environmental dangers.

*For correspondence: john.
carlson@yale.edu

**Competing interests:** The authors declare that no competing interests exist.

## Introduction

The sense of taste allows animals to identify food sources that are nutritious but not toxic (*Liman et al., 2014*). Many toxins elicit a bitter taste and an aversive response that is widely conserved across phyla. Bitter-tasting compounds are of diverse structures and chemical classes. Many bitter compounds are synthesized by plants and deter insects from feeding on them (*Pontes et al., 2014*; *Briscoe et al., 2013*; *Salloum et al., 2011*; *Sellier et al., 2011*; *Freeman and Dahanukar, 2015*). Insects in turn have evolved gustatory systems that detect these compounds and signal the danger of toxicity (*Wada-Katsumata et al., 2013*).

The principal taste organ of the *Drosophila* head is the labellum (*Figure 1*). The labellum contains ~31 taste sensilla that fall into morphological classes based on length: short (S; green and blue in *Figure 1*), intermediate (I; purple and red), and long (L; gray in *Figure 1*) (*Stocker, 1994*). Each sensillum has a pore at its tip. When a sensillum makes contact with a potential food source, compounds from the food source diffuse through the pore and activate gustatory receptor neurons inside.

Electrophysiological analysis of all 31 sensilla with a panel of 16 bitter compounds revealed that bitter compounds elicit different responses from different sensilla (*Weiss et al., 2011*). Four functional classes of bitter-sensitive sensilla were identified, two of short length, S-a and S-b, and two of intermediate length, I-a and I-b. Each class contains a neuron excited by bitter compounds, referred to as a 'bitter neuron,' with a distinguishable response spectrum.

Expression analysis of all 68 members of the Gustatory receptor (Gr) family identified four corresponding classes of bitter neurons, each expressing a distinct subset of receptors (*Figure 1*) (*Weiss et al., 2011*). The receptors expressed in bitter neurons are referred to for convenience as 'bitter receptors.' They show extensive coexpression: 29 are coexpressed in the bitter neuron of S-a; 16 in S-b; 6 in I-a; and 10 in I-b. In addition to the labellum, the legs and pharynx of the fly also have gustatory function, as does the larval head (*Weiss et al., 2011*; *Kwon et al., 2011*; *LeDue et al., 2015*; *Ling et al., 2014*; *Meunier et al., 2003*; *Oppliger et al., 2000*). We note that expression

**eLife digest** Insects and other animals use their sense of taste to tell if their food is safe to eat. Plant toxins, for example, often have a bitter flavor that animals can detect and avoid. Fruit flies have many bitter-sensitive nerve cells, but it is not known how the receptors on these nerve cells signal the detection of bitter-flavored compounds.

Delventhal and Carlson have now used fruit flies to investigate how taste receptors of the so-called Gustatory receptor family detect bitter flavors. The experimental approach involved genetically modifying four different types of nerve cells that sense bitter compounds so that they produced higher levels of particular taste receptors than normal. Then, the flies were exposed to a range of bitter compounds while the electrical activity of each cell was measured.

The analysis involved about 600 combinations of receptors, nerve cells and compounds. In some bitter-sensing nerve cells, increasing the number of taste receptors increased the cell's responsiveness to bitter compounds. However, in other nerve cells, similar modifications suppressed an existing response or resulted in a new response.

Delventhal and Carlson propose that these results suggest the specific response of a bitter-sensing nerve cell depends on the interactions between its different taste receptors. Furthermore, the ability of receptors to compete, inhibit or activate each other in different ways could have implications for evolution. For example, such flexible interactions might allow a taste system to evolve new, enhanced or diminished responses to new food sources and tastes in a changing environment. It now remains to be investigated how such receptor interactions take place at a molecular level.

analysis of Grs has been based primarily on *Gr-GAL4* drivers, since success in analyzing Gr expression with in situ hybridization has been extremely limited.

Loss-of-function studies have clearly shown that some Grs are required for the responses to certain bitter compounds. For example, *Gr93a* is required for the behavioral and physiological response to caffeine; *Gr8a* is required for response to L-canavanine; *Gr66a, Gr33a* and *Gr32a* are broadly required for the response to several bitter tastants (*Lee et al., 2009*; *2010*; *2012*; *Moon et al., 2006*; *Moon et al., 2009*). However, the roles of individual Grs in bitter response have been difficult to discern in detail, in part because of the coexpression of bitter receptors and in part because of the difficulty of expressing bitter receptors in conventional expression systems such as cultured cells or oocytes (*Liman et al., 2014*).

In vivo expression studies provide a complementary approach for analysis of Gr function. Here, we analyze eight bitter Grs, most of which have not been functionally studied before. We express them in a natural laboratory: in bitter neurons of the labellum that either do or do not express them endogenously, which we refer to as overexpression or ectopic expression, respectively. Specifically, we express individual Grs in four different bitter neurons, three of wild type and one of a *Gr* mutant, and we measure the effects of their expression using electrophysiology and a panel of 21 bitter tastants. From an analysis of ~600 receptor-neuron-tastant combinations ($n \geq 27,000$ total recordings) we find several surprising results. While expression of Grs led to increases in many responses, expression of some Grs led to decreases in certain responses. Conversely, the deletion of one *Gr* from a neuron led to novel responses not observed in wild type. Overexpression of a Gr in some neurons led not only to increased responses but also to novel responses. A recurrent theme was that the expression of the same Gr in different neurons led to strikingly different results. Taken together, our findings provide support for a model in which bitter Grs interact, exhibiting competition, inhibition, or activation in different contexts. The results may have profound evolutionary implications: they suggest a rich source of means by which the taste system can evolve novel, increased, or decreased responses to new environmental opportunities and dangers, or modulate its response to accommodate changes in the internal state of the fly.

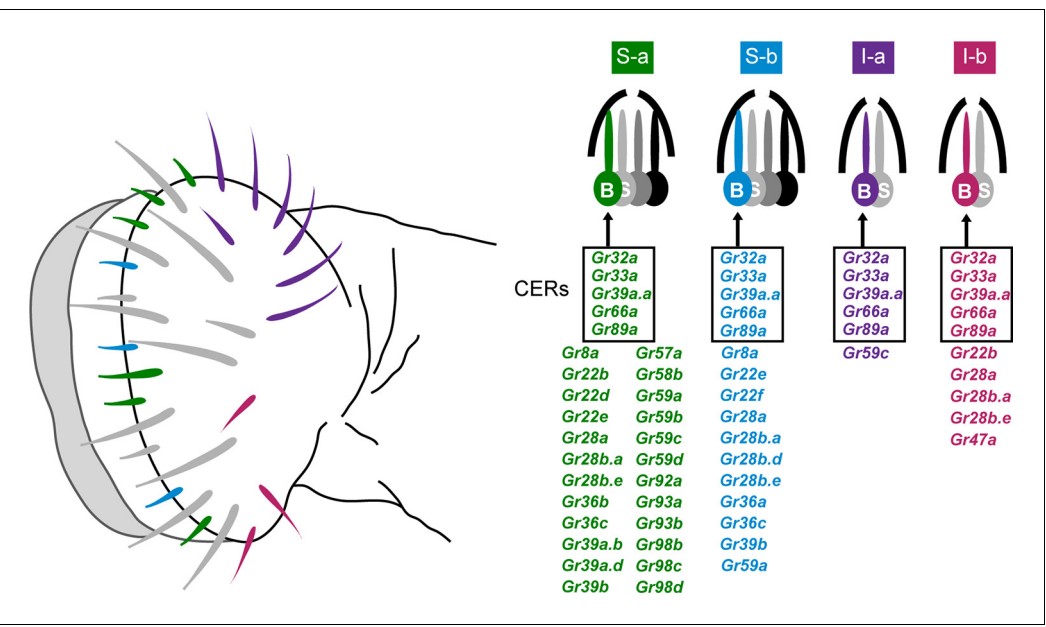

**Figure 1.** Sensillum types on the labellum. Left: The four bitter-responsive types, S-a (green), S-b (blue), I-a (purple) and I-b (red), are differently distributed on the labellar surface. L sensilla (gray) show little if any response to bitter compounds. Right: S-a and S-b have four gustatory receptor neurons (GRNs), one of which is bitter-responsive, and I-a and I-b sensilla have two GRNs, one of which is bitter-responsive. The bitter-responsive neuron (B) of each sensillum type expresses a different combination of Grs. Five Grs, referred to as 'Commonly Expressed Receptors' (CERs; in rectangles), are expressed in every bitter neuron on the labellum. Many or all sensilla also contain a sugar-sensitive neuron (S). Mapping of Grs to neurons is based on *GAL4* driver expression. Figure adapted from *Weiss et al. (2011)*.

## Results

### Novel responses conferred by ectopic expression of Grs in I-a bitter neurons

To investigate the function of bitter Grs, we ectopically expressed them in bitter neurons of the labellum via the *GAL4-UAS* system (*Brand and Perrimon, 1993*). We then measured the effects of this expression via electrophysiology (*Benton and Dahanukar, 2011*; *Delventhal et al., 2014*). The aim of this approach was to study bitter Grs in an environment similar to their native environment— that is, in bitter taste neurons, in taste sensilla, in vivo.

We initially expressed Grs in bitter neurons of I-a sensilla (*Figure 1*, purple sensilla). These neurons were selected because they have the narrowest response spectrum of the various classes of labellar bitter neurons (*Weiss et al., 2011*). These sensilla also express the smallest number of bitter *Gr* genes, as determined in a systematic *Gr-GAL4* expression analysis (*Figure 1*).

As an initial *Gr* gene we examined *Gr22b*, which in wild-type flies is expressed in S-a sensilla, but not I-a sensilla. We used *Gr89a-GAL4,* which drives expression in all labellar bitter neurons, and then measured responses of I-a sensilla to a broad panel of bitter tastants. We found that ectopic expression of *Gr22b* conferred an electrophysiological response to cucurbitacin (CUC) that is not observed in either of the parental control lines, *GAL89a-GAL4; +* or *+/UAS-Gr22b* (*Figure 2a, b*). CUC is a plant-derived compound that has insecticidal activity and tastes bitter to humans (*Torkey et al., 2009*). Expression of Gr22b also conferred strong responses to sucrose octaacetate (SOA), another bitter-tasting compound (*Harwood et al., 2012*), and to azadirachtin (AZA), which is found in the seeds of the neem tree and which inhibits feeding of locusts and many other insects (*Table 1*) (*Schmutterer, 1990*; *Sharma et al., 1993*).

To confirm and extend these results, we tested responses to SOA across a range of concentrations (*Figure 2c*). Responses conferred by Gr22b increased above 0.1 mM and appeared to saturate at 20 spikes/s. Neither of the parental control lines responded at any concentration.

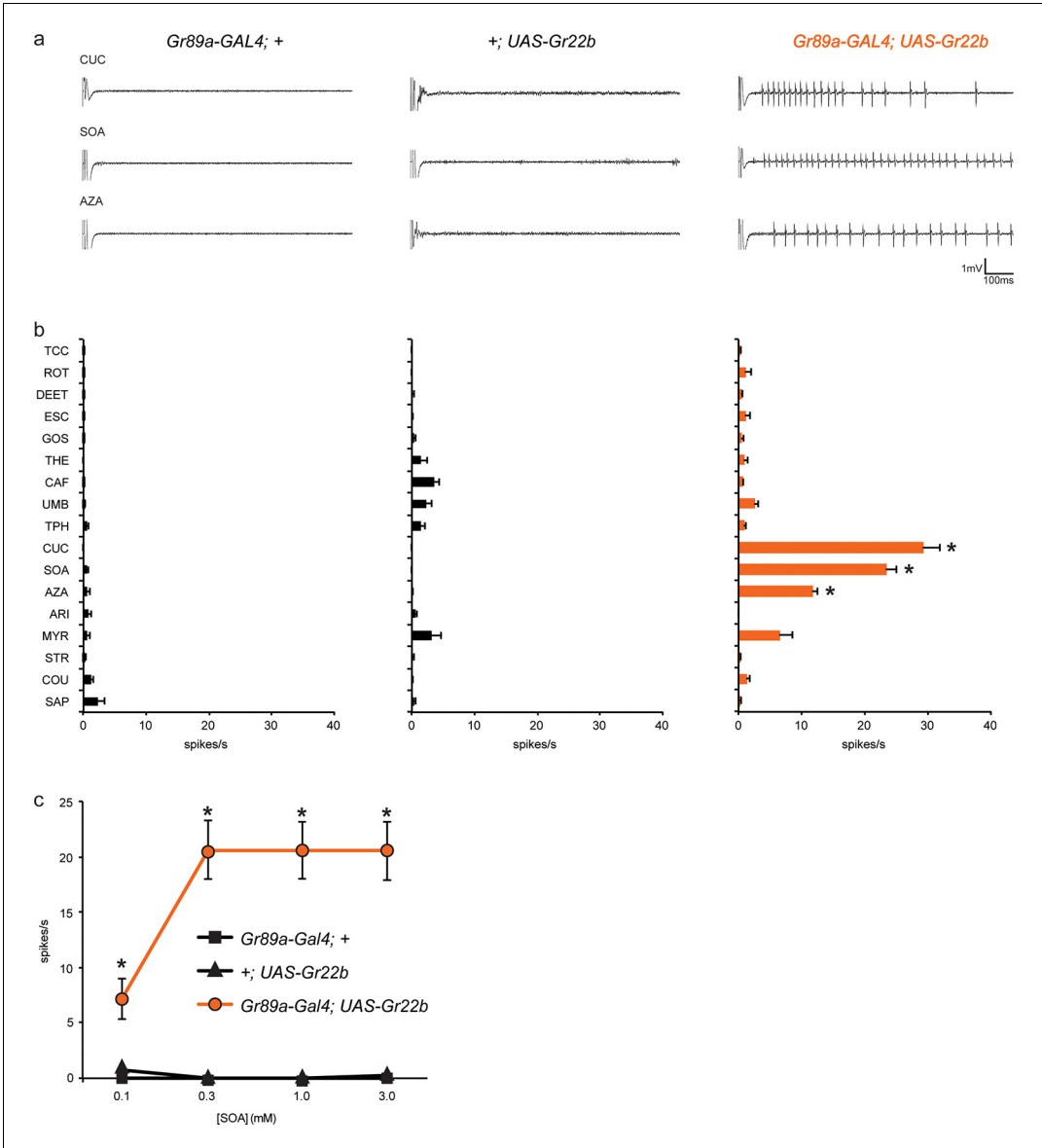

**Figure 2.** Ectopic expression of *Gr22b* in I-a bitter neurons leads to novel responses to three bitter compounds. (a) Sample electrophysiological responses from I-a sensilla of parental controls and of flies ectopically expressing *UAS-Gr22b*. (b) Mean responses. Asterisks indicate responses that are different from both parental controls as measured by a two-way ANOVA, with Bonferroni multiple comparisons correction (p ≤ 0.0001, n ≥ 20). Only compounds that do not elicit responses greater than that from a TCC control solution in wild type I-a sensilla, as measured by a one-way ANOVA, are shown. (c) Responses of I-a sensilla to SOA in *Gr22b*-expressing flies, across a range of concentrations, relative to both parental controls (* indicates p ≤ 0.0001, n ≥ 17). Concentrations are graphed on a logarithmic scale.

Taken together, these results indicate that expression of Gr22b confers novel responses in a neuron that does not normally express it. We note that CUC, SOA, and AZA all elicit responses from wild type S-a sensilla, which express Gr22b endogenously.

Next, we asked whether a number of other Grs might also confer responses to I-a, and if so, whether the responses differed. Six additional Grs were chosen to analyze in detail, including Grs expressed in the I-b, S-a, and S-b classes of bitter-sensitive labellar sensilla, in bitter-sensitive leg sensilla, in the pharynx, in the larva, and in the adult antenna (*Table 2*) (*Weiss et al., 2011*; *Kwon et al., 2011*; *2014*; *Scott et al., 2001*; *Fishilevich and Vosshall, 2005*). We expressed each

**Table 1.** Panel of 21 bitter taste compounds tested in electrophysiological recordings. [a] '−' indicates that insecticidal activity has not been described.

| Tastant | Abbreviation | Concentration | Chemical class | Source | Insecticidal activity[a] |
|---|---|---|---|---|---|
| Aristolochic acid | ARI | 1 mM | phenanthrene | *Aristolochia* family of plants | − |
| Azadirachtin | AZA | 1 mM | terpenoid | Neem tree | + |
| Berberine chloride | BER | 1 mM | alkaloid | Golden seal, bayberry, Oregon grape and goldthread | − |
| Caffeine | CAF | 1 mM | alkaloid | Coffee, chocolate, tea, kola nut | − |
| Coumarin | COU | 10 mM | benzopyrone | Tonka bean, honey clover | + |
| Cucurbitacin I hydrate | CUC | 1 mM | glycoside | Pumpkins, gourds, cucumbers | + |
| N,N-Diethyl-m-toluamide | DEET | 10 mM | N,N-dialkylamide | synthetic | + |
| Denatonium benzoate | DEN | 10 mM | quaternary ammonium cation | synthetic | − |
| Escin | ESC | 1 mM | terpenoid | Horse chestnut tree | − |
| Gossypol | GOS | 1 mM | terpenoid | Cotton | + |
| (-)-lobeline HCl | LOB | 1 mM | alkaloid | Indian tobacco, Cardinal flower | + |
| Myricetin | MYR | 1 mM | flavonoid | Berries, wine | − |
| Quinine | QUI | 1mM | alkaloid | Cinchona tree bark | − |
| Rotenone | ROT | 1 mM | ketone | Jicama | + |
| Saponin | SAP | 1% | terpenoid | Soapbark tree | + |
| D-(+)-sucrose octaacetate | SOA | 1 mM | acetylated sucrose derivative | synthetic | + |
| Sparteine sulfate salt | SPS | 10 mM | alkaloid | Scotch broom | + |
| Strychnine nitrate salt | STR | 10 mM | alkaloid | *Strychnos* seeds | + |
| Theobromine | THE | 1 mM | alkaloid | Cacao, tea, kola nut, chocolate | − |
| Theophylline | TPH | 10 mM | alkaloid | Tea leaves | + |
| Umbelliferone | UMB | 1 mM | phenylpropanoid | Carrot, coriander | − |

of these receptors individually using the *Gr89a-GAL4* driver and measured the responses to an expanded panel of 21 chemically diverse bitter compounds in I-a sensilla. As part of this analysis we extended our analysis of Gr22b to include the entire tastant panel.

Four of these seven Grs conferred responses to three or more compounds that elicited no response from the control line, referred to henceforth as novel responses ('N' in *Figure 3a*). Gr22b, in addition to conferring the novel responses shown in *Figure 2b*, also conferred greater responses to DEN, BER, and LOB than the control ($p < 0.0001$, two-way ANOVA, Bonferroni test, $n \geq 20$).

Gr58c conferred novel responses to MYR, STR, and COU, and responses to QUI, LOB, and DEN that were greater than those of the control line ($p < 0.0001$, except that for COU $p < 0.0005$, and for MYR $p < 0.05$, $n \geq 16$). We note that Gr58c is expressed in leg sensilla that respond to five of these compounds; the sixth was not tested in legs (*Ling et al., 2014*).

Remarkably similar response profiles were conferred by expression of Gr2a and Gr10a. Both receptors conferred responses to caffeine (CAF), umbelliferone (UMB), and theophylline (TPH), which elicit essentially no response from the control, and an increased response to LOB ($p < 0.0001$, except $p < 0.01$ for the TPH values and $p < 0.05$ for CAF in the case of Gr10a, $n \geq 8$). Despite this functional similarity, Gr2a and Gr10a are distantly related phylogenetically (*Robertson et al., 2003*). CAF and TPH, which are present in coffee and tea, are closely related in structure; UMB is present in carrots and coriander and is structurally distinct (*Table 1*).

In an independent dose-response experiment, the response to UMB increased above 0.3 mM, and appeared to saturate at a level between 10 spikes/s and 15 spikes/s (*Figure 3b*; limited

**Table 2.** Endogenous expression patterns of Grs selected for analysis, as determined primarily by *Gr-GAL4* analysis.

| Gene | Labellum | Legs | Pharynx | Larva | Antenna |
|------|----------|------|---------|-------|---------|
| Gr2a | — | — | + | + | — |
| Gr10a | — | — | — | + | + |
| Gr22b | + (I-b, S-a) | + | + | + | — |
| Gr28a | + (I-b, S-a, S-b) | + | + | + | — |
| Gr28b.a | + (I-b, S-a, S-b) | + | + | + | — |
| Gr36a | + (S-b) | + | — | — | — |
| Gr58c | — | + | — | — | — |
| Gr59c | + (I-a, S-a) | — | — | + | — |

solubility of UMB precluded testing at higher concentrations). Neither parental control responded to any tested concentration of UMB. We note that saturation of the response to UMB conferred by Gr2a was lower than that for the response to SOA conferred by Gr22b (*Figure 2c*); the SOA response also appeared to have a lower threshold and to saturate at a lower concentration than the response to UMB.

When Gr28b.a was expressed, we observed a modest response to MYR that was not observed in the control line ($p < 0.05$, $n \geq 26$), as well as an increased response to LOB ($p < 0.0001$, $n \geq 26$). Gr28a produced an increase in response to LOB ($p < 0.0001$, $n \geq 24$). Gr36a produced no changes in the response profile of I-a. *Gr36a* could require a co-factor not present in I-a, or could function in response to a tastant not included in the panel.

## Different effects of Gr expression in different neurons

We asked whether the three receptors that conferred the most modest effects, or no effects, to I-a sensilla, *i.e.* Gr28b.a, Gr28a, and Gr36a, behaved similarly in other sensilla. Again using the *Gr89a-GAL4* driver, we measured the effect of expressing them on the responses of the bitter neurons of I-b and S-a sensilla.

Expression of Gr28b.a in I-b sensilla, where it is expressed endogenously, conferred an increased response to aristolochic acid (ARI), as well as to berberine (BER) (*Figure 4a*). The increase to ARI was confirmed in an independent dose-response analysis: responses in I-b sensilla of the *Gr89a-GAL4; UAS-Gr28b.a* line were greater than in either parental control at the two higher concentrations tested (*Figure 4b*; higher concentrations were not tested due to limited solubility). At the highest concentration, the response of *Gr89a-GAL4; UAS-Gr28b.a* approached 25 spikes/s.

Expression of Gr28a in I-b sensilla conferred a novel response to saponin (SAP) not observed in the *GAL4* control line (*Figure 4a*); the response was not observed in the *UAS-Gr28a* parental control either (1 spike/s ± 1, n=11). The production of this novel response was surprising, since Gr28a is expressed endogenously in I-b sensilla. Gr36a conferred no response to I-b, as in I-a.

Thus the effects of expression of Gr28b.a and Gr28a in I-b differ from those in I-a. In contrast to I-b, no responses to ARI or SAP were conferred by expression of Gr28b.a or Gr28a, respectively, in I-a sensilla. Conversely, the increases in response to MYR and LOB that were observed in I-a were not observed in I-b. The simplest interpretation of these results, taken together, is that some Grs have distinct functions in different neuronal contexts.

We also expressed the three Grs in S-a sensilla. No increases in response were observed to any tastant, with any receptor. These results lend further support to the conclusion that the effects of Gr expression vary in different neuronal contexts. However, a striking effect was observed in the response to DEN, UMB, ARI, and BER upon expression of *UAS-Gr28a* in S-a: responses were lower

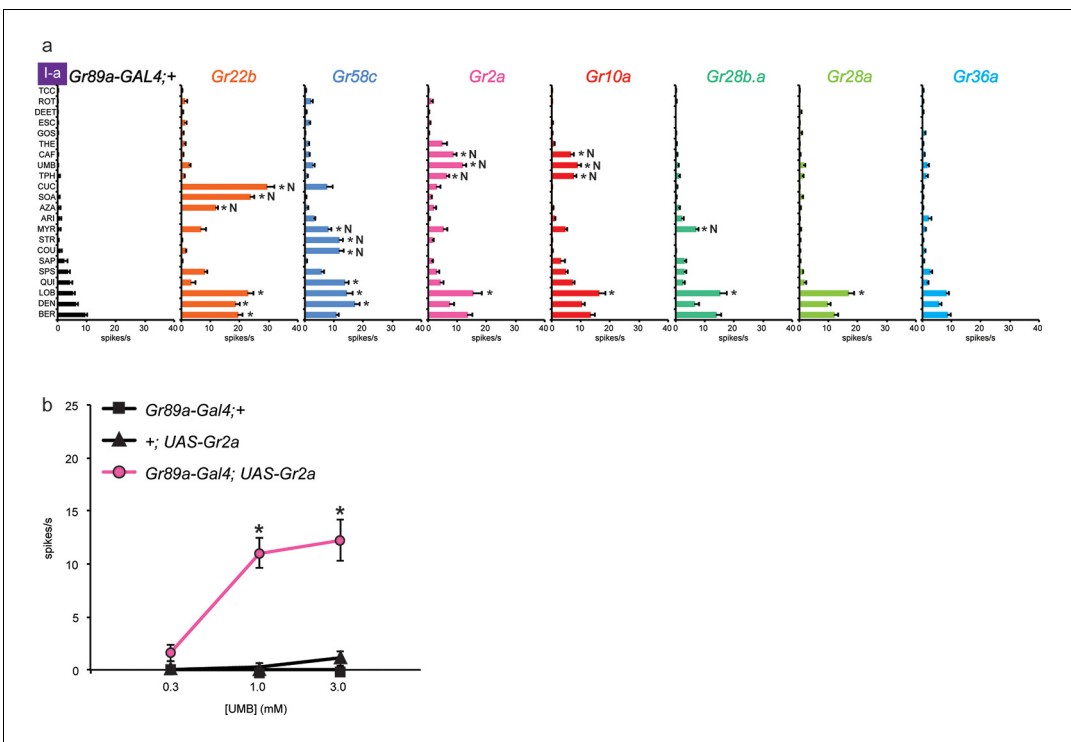

**Figure 3.** Electrophysiological responses of sensilla in which seven individual Grs are expressed in I-a bitter neurons. (**a**) 'N' indicates a novel response, to which there was no significant response in the wild-type control. Each experimental genotype is *Gr89a-GAL4; UAS-GrX*. Asterisks indicate responses that are different from the *Gr89a-GAL4;+* parental control (two-way ANOVA, with Bonferroni multiple comparisons correction. $p \leq 0.0001$, except that $p \leq 0.001$ for Gr58c/COU; $p<0.01$ for Gr2a/TPH and Gr10a/TPH; $p \leq 0.05$ for Gr58c/MYR, Gr10a/CAF, and Gr28b.a/MYR. $n \geq 7$). (**b**) Responses to UMB from Gr2a-expressing flies, across a range of concentrations, and relative to both parental controls (asterisks indicate $p \leq 0.0001$, $n \geq 17$). Concentrations are graphed on a logarithmic scale.

than in the parental control line (*Figure 4c*). A similar finding is presented in the next section, where it is considered in more detail.

## Suppression of endogenous responses by expressed Grs

We asked whether the Grs that conferred responses to three or more compounds in I-a (Gr22b, Gr58c, Gr2a and Gr10a) produced similar effects in other sensilla. We first tested the effects of Gr22b expression in S-a sensilla. As in I-a, Gr22b expression conferred an increased response to SOA and CUC (*Figure 5a*). The *Gr89a-GAL4; UAS-Gr22b* line also responded strongly to AZA, as in I-a, but the responses of the control lines were equally strong.

Dramatic reductions, however, were produced by Gr22b expression in the responses of S-a to a number of tastants, such as DEN (*Figure 5a*). This decreased DEN response in S-a is in contrast to the increased DEN response in I-a produced by expression of the same receptor (*Figure 3a*). This suppression is particularly interesting because Gr22b is expressed endogenously in S-a sensilla; thus suppression of response is observed in a cell that has been engineered to overexpress one of its endogenous receptors.

Gr58c also suppressed the responses of S-a sensilla, in this case to UMB and ARI (*Figure 5b*). Gr58c did not suppress the response to DEN; thus Gr58c and Gr22b both suppress responses to bitter compounds, but different compounds. Gr2a reduced responses in S-a to several tastants, including DEN, SPS, STR, and LOB. Gr10a also showed suppressive effects in S-a. Moreover, the profiles produced by Gr2a and Gr10a were very similar to each other (*Figure 5b*), as in I-a (*Figure 3a*).

We confirmed the suppression of S-a responses via an independent dose-response analysis (*Figure 5c*). At all concentrations tested, responses to SPS were lower in the line expressing Gr2a

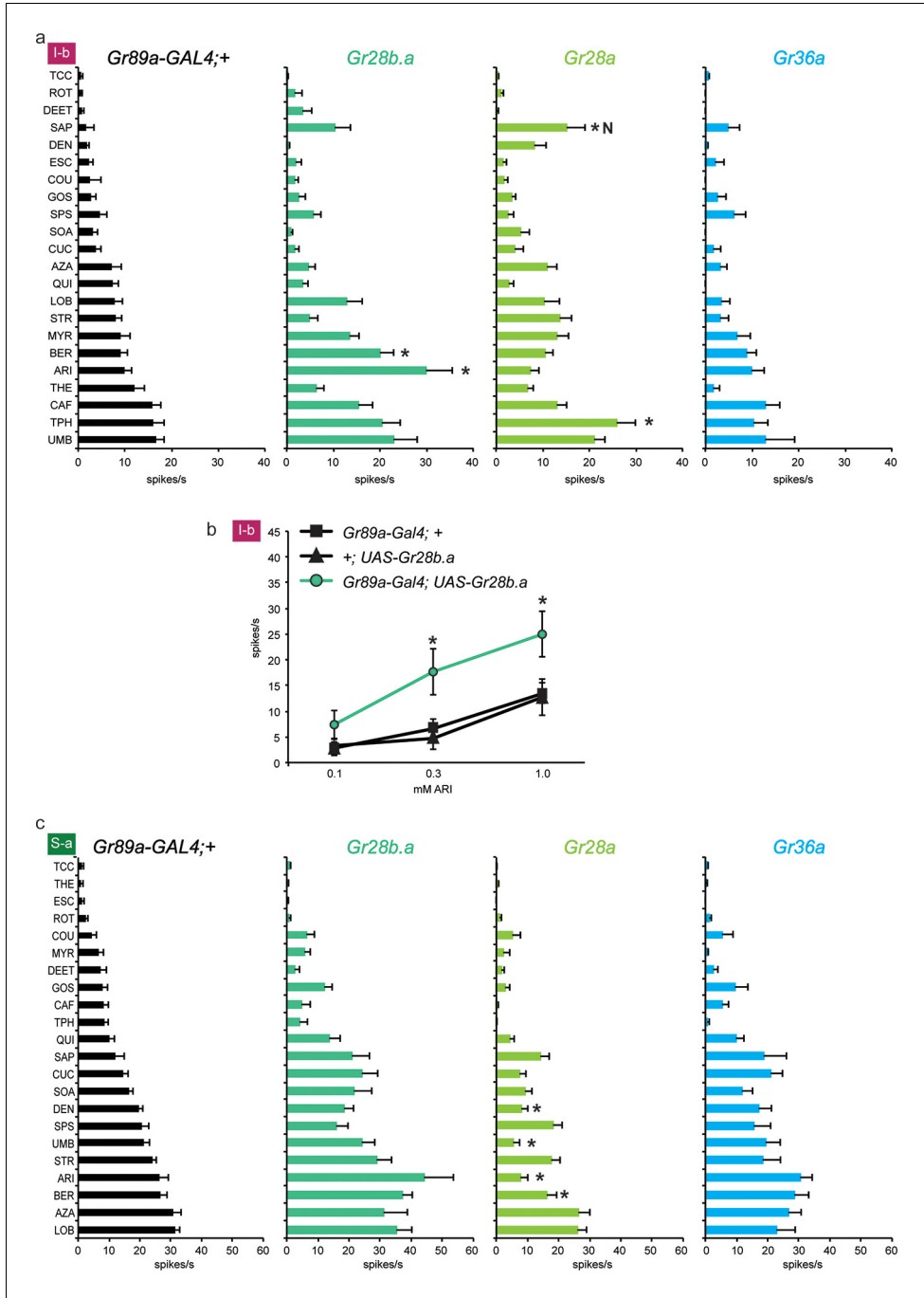

**Figure 4.** Electrophysiological responses of sensilla in which three individual Grs (Gr28b.a, Gr28a, Gr36a) are expressed in I-b (a) and S-a (c) bitter neurons. Tastant order and x-axis scales differ between panels a and c for clarity of presentation. The experimental genotypes were *Gr89a-GAL4; UAS-GrX*. (a) In I-b sensilla, Gr28b.a conferred an increased response to ARI ($p \leq 0.0001$, $n \geq 10$) and BER ($p \leq 0.01$, $n \geq 10$) relative to the *GAL4* parental control line. Gr28a conferred response to SAP ($p \leq 0.001$, $n \geq 13$) and TPH ($p \leq 0.05$, $n \geq 13$) relative to the *GAL4* parental control line. Gr36a conferred no increased responses ($n \geq 6$). (b) A dose-response analysis using both parental controls revealed increases in ARI response in I-b sensilla (* indicates $p \leq 0.05$, $n \geq 22$). Concentrations are graphed on a logarithmic scale. (c) In S-a sensilla, Gr28a conferred decreased responses (ARI: $p \leq 0.0001$, UMB: $p \leq 0.001$, DEN and BER: $p \leq 0.02$. $n \geq 11$), while Gr28b.a and Gr36a did not ($n \geq 6$).

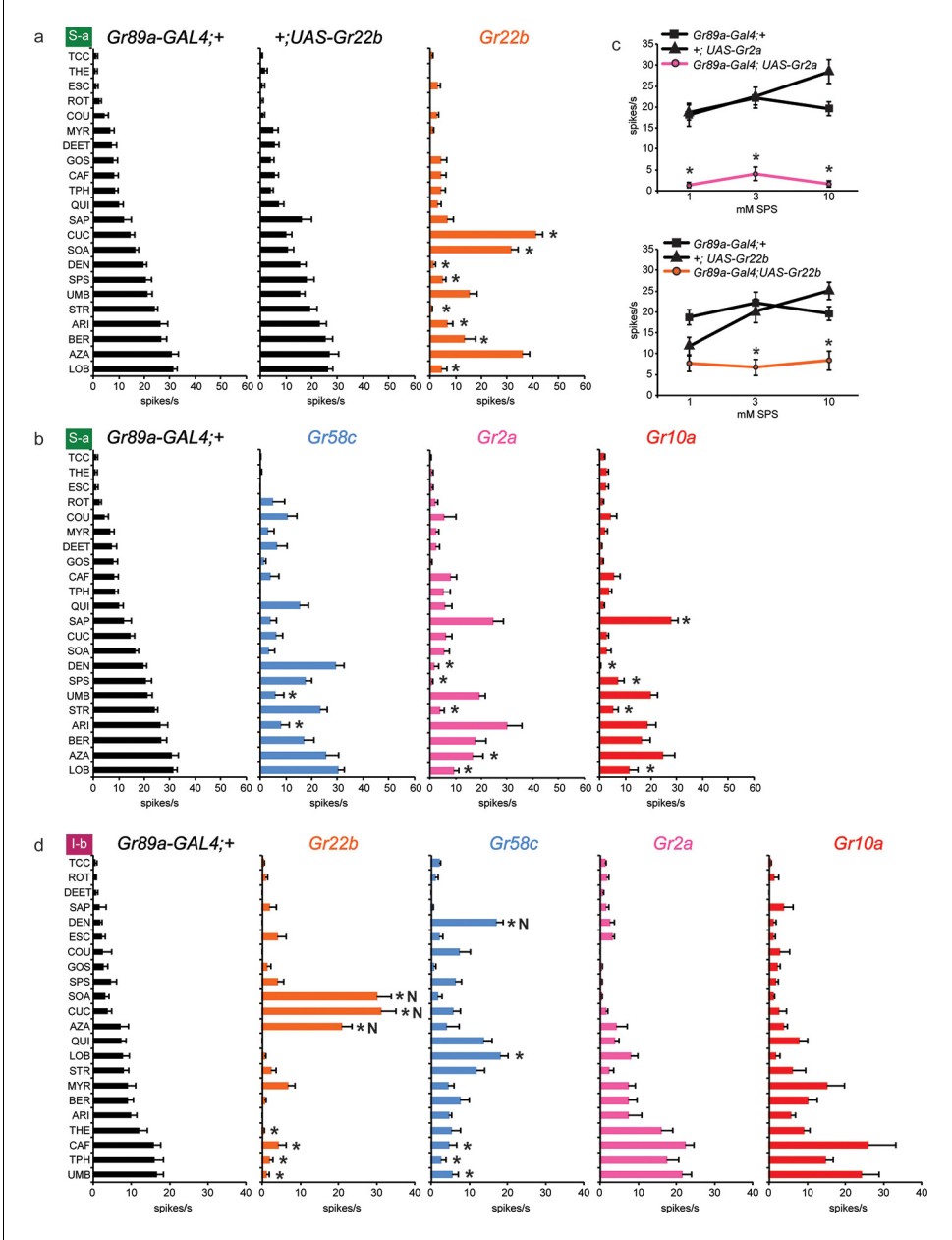

**Figure 5.** Electrophysiological responses of sensilla in which four individual Grs are expressed in S-a and I-b bitter neurons. Each experimental genotype is *Gr89a-GAL4; UAS-GrX*. (a) Response profiles of both parental controls and flies expressing *UAS-Gr22b* in S-a bitter neurons. Asterisks indicate responses that are different from both parental controls (two-way ANOVA, with Bonferroni multiple comparisons correction; p ≤ 0.0001, except BER: p ≤ 0.0005. n ≥ 7). Response profiles generated by expression of Gr22b, Gr58c, Gr2a, Gr10a in S-a bitter neurons (b) and I-b bitter neurons (d). Asterisks indicate responses that are different from the parental control (two-way ANOVA, with Bonferroni multiple comparisons correction; p ≤ 0.0001, except that for S-a sensilla, p≤0.001 for Gr2a/SPS, Gr2a/LOB, Gr10a/DEN, Gr10a/LOB; p ≤ 0.01 for Gr2a/DEN, Gr2a/AZA, Gr58c/ARI; p≤0.03 for Gr10a/SAP, Gr10a/SPS, and Gr58c/UMB. For I-b sensilla, p ≤ 0.001 for Gr22b/TPH; p≤0.01 for Gr22b/THE, Gr22b/CAF, Gr58c/LOB, and Gr58c/TPH; p ≤ 0.05 for Gr58c/CAF and Gr58c/UMB. n ≥ 6). (c) Expression of Gr2a and Gr22b in S-a bitter neurons conferred suppression of the endogenous response to SPS. Asterisks indicate responses that are different from both parental controls (p ≤ 0.002, n ≥ 18). Concentrations are graphed on a logarithmic scale.

than in the parental control lines. Suppression of SPS response by Gr22b was observed at both 3 mM and 10 mM concentrations.

Suppression of responses was observed not only in S-a sensilla but also in I-b sensilla (*Figure 5d*). Although Gr22b expression conferred novel responses to SOA, CUC and AZA, as it did in I-a sensilla (*Figure 3a*), it suppressed the responses to THE, CAF, TPH, and UMB. The ability of Gr22b to confer entirely novel responses and suppress endogenous ones in I-b sensilla is especially surprising given that Gr22b is endogenously expressed in I-b bitter neurons. Likewise, Gr58c showed reduced responses to CAF, TPH, and UMB, while conferring response to DEN and LOB, two compounds to which Gr58c expression also confers responses in I-a sensilla (*Figure 3a*). Gr2a and Gr10a again behaved similarly in I-b sensilla, in that their expression had no effects. Their lack of effect in I-b sensilla is in contrast to their conferral of responses in I-a sensilla and their suppression of responses in S-a sensilla.

In summary, suppression occurred broadly: in both S-a and I-b, by several different receptors, and with several different tastants. Expression of an individual receptor in a given sensillum increased the responses to some tastants and decreased the responses to others (e.g. Gr22b; CUC and DEN in S-a). Expression of an individual receptor increased the response to a tastant in one sensillum type and decreased it in another (e.g. Gr22b; DEN in I-a and S-a). Suppression was observed both in cases of overexpression and ectopic expression, i.e. expression of a receptor in a neuron that either does (e.g. Gr22b in S-a) or does not (e.g. Gr2a in S-a) express it endogenously in wild type. Finally, in some sensilla a receptor produced decreased responses but no increased responses (e.g. Gr2a in S-a).

## Deletion of *Gr59c* converts the I-a profile to one similar to that of I-b

The foregoing analysis concerned expression of Grs that are coexpressed in labellar bitter neurons with many other Grs, or that are not expressed in the labellum. We next considered Gr59c, which is the sole receptor in the I-a bitter neurons other than the commonly expressed receptors (CERs) (*Figure 1*); it is also expressed in S-a but not I-b. We analyzed it through both in vivo expression analysis and loss-of-function analysis.

First, we drove its expression in I-b, S-a, and I-a sensilla. A previous test with a limited number of tastants found that misexpression of Gr59c increased or decreased the responses to certain tastants (*Weiss et al., 2011*). Using our complete panel of 21 tastants we confirmed and extended these results: we found that expression of Gr59c increased responses to LOB, DEN and BER in all three sensillum types, and suppressed many of the larger responses in I-b and S-a (*Figure 6*). Interestingly, the net result in each sensillum type was a profile in which LOB, DEN, and BER elicited the strongest responses.

Second, we constructed a deletion of *Gr59c* (△*Gr59c*) (*Figure 7—figure supplement 1*). We hypothesized that since Gr59c expression confers a response to LOB, DEN, and BER, and since these three tastants elicit the greatest mean responses from I-a among all the tastants in the panel, that a △*Gr59c* I-a sensillum might display weak or no responses to all tastants of the panel. If so, moreover, it might provide a particularly useful in vivo system in which to express other Grs.

As expected, deletion of *Gr59c* eliminated the responses to LOB, DEN, and BER (*Figure 7*). However, contrary to our expectations, △*Gr59c* I-a sensilla showed strong novel responses to THE, CAF, UMB, and TPH. Intriguingly, the response profile of the mutant I-a sensillum is similar to that of control I-b sensilla (*Figure 7*). Expression of a *UAS-Gr59c* construct in △*Gr59c* flies was sufficient to restore a profile similar to that of wild-type I-a sensilla, providing evidence that the mutant phenotype is in fact due to loss of *Gr59c*. We note that this rescue argues against the notion that the flanking *Gr59d* gene, which is also removed by the *Gr59c* deletion (*Figure 7—figure supplement 1*), contributes to the △*Gr59c* phenotype. Moreover, a *GAL4-Gr59d* driver shows expression in S-a bitter taste neurons but not in I-a sensilla, consistent with the interpretation that the △*Gr59c* phenotype observed in I-a sensilla arises from loss of *Gr59c* alone (*Weiss et al., 2011*).

We asked whether △*Gr59c* has a similar effect on the S-a bitter neuron, which also expresses *Gr59c*. We found no effect, other than a decrease in the response to DEN (*Figure 7—figure supplement 2*, p<0.0001, n≥12). The lack of a strong mutant phenotype may reflect the much greater molecular complexity of S-a: apart from the CERs, the S-a bitter neuron expresses 24 Grs, whereas its I-a counterpart expresses only Gr59c. The difference between the △*Gr59c* phenotype in the two

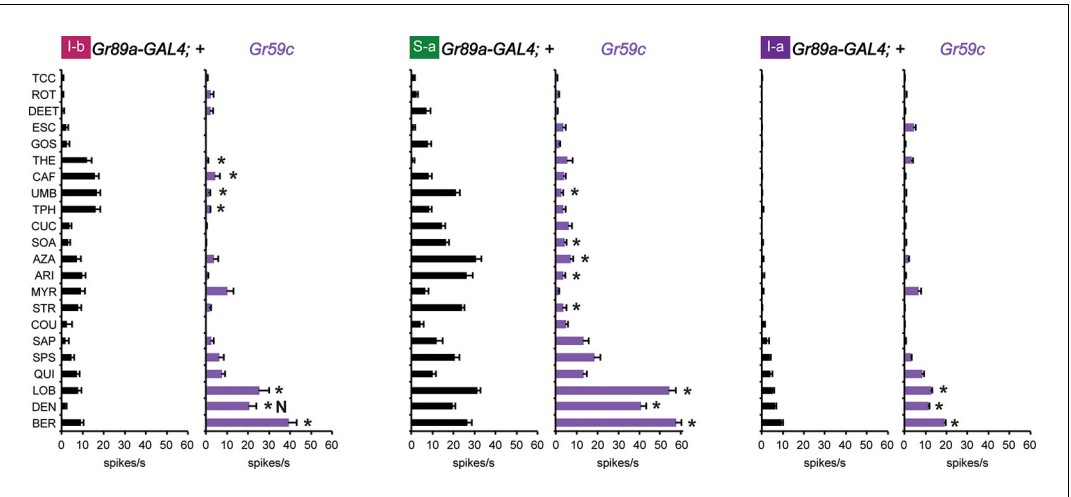

**Figure 6.** Electrophysiological response profiles generated by expression of Gr59c in I-b, S-a, and I-a, relative to the wild-type *GAL4* parental control (p ≤ 0.001, except that p≤ 0.05 in the case of the response of I-b to THE and CAF. n ≥ 8). The experimental genotype was *Gr89a-GAL4; UAS-Gr59c*.

sensilla provides a further example of how the function of a bitter Gr may differ in different neuronal contexts.

## The role of *Gr59c* in the phenotypic switch

How might deletion of *Gr59c* cause an I-a bitter neuron to shift its response profile, particularly to one similar to that of an I-b neuron? One possibility is that loss of *Gr59c* changes the developmental fate of I-a bitter neurons. We addressed this possibility in two ways, through a molecular analysis and through a developmental genetic analysis.

We first considered the hypothesis that deletion of *Gr59c* induces expression of other Grs in the I-a sensillum. Our interest in this hypothesis was motivated by the finding that mutation of a green light-sensing receptor in the *Drosophila* visual system, Rh6, leads to the ectopic expression of a blue-sensitive receptor, Rh5 (*Vasiliauskas et al., 2011*). Likewise, deletion of a mammalian odor receptor gene leads to the expression of another odor receptor (*Lomvardas et al., 2006*; *Magklara and Lomvardas, 2013*).

We tested the expression of four *Gr-GAL4* drivers, one of which is expressed in wild-type I-a sensilla but not I-b sensilla (*Gr59c-GAL4*), and three of which are not expressed in I-a but are expressed in I-b (*Gr28a-, Gr28b.a-* and *Gr22b-GAL4*) (*Figures 1*, *8*; arrowheads indicate the expected positions of cell bodies of representative I-a sensilla). All drivers were tested in both △*Gr59c* and control backgrounds. If the I-a bitter neurons underwent a complete change in cell fate to that of I-b bitter neurons, then one would expect these neurons to lose expression of *Gr59c-GAL4*. This outcome was not observed (*Figure 8*, top two panels). In fact, expression of *Gr59c-GAL4* appeared the same in the △*Gr59c* labellum as in the control. Likewise, none of the three drivers that are expressed in I-b were expressed in △*Gr59c* I-a sensilla (*Figure 8*, bottom six panels). These results argue against the hypothesis that I-a sensilla undergo a complete fate change switch to I-b sensilla.

We also tested the hypothesis that *Gr59c* plays a role during development in determining the fate of taste neurons. Using a △*Gr59c* mutant background, we restored *Gr59c* expression at different times, during and after development. Specifically, we introduced a *UAS-Gr59c* rescue construct under the control of the *Gr89a-GAL4* driver, but we repressed the driver with a temperature-sensitive *GAL80$^{ts}$* construct (*Lee and Luo, 1998*; *Zeidler et al., 2004*). We increased the temperature at varying times, thereby inactivating *GAL80$^{ts}$*, relieving the repression, and turning on *Gr59c*. We asked whether the I-a sensilla had a wild-type phenotype (responses to BER, LOB, and DEN) or a mutant phenotype (responses to UMB, TPH, and CAF). The flies were examined at 7-9d, by testing the physiological responses of sensilla to a diagnostic panel of tastants.

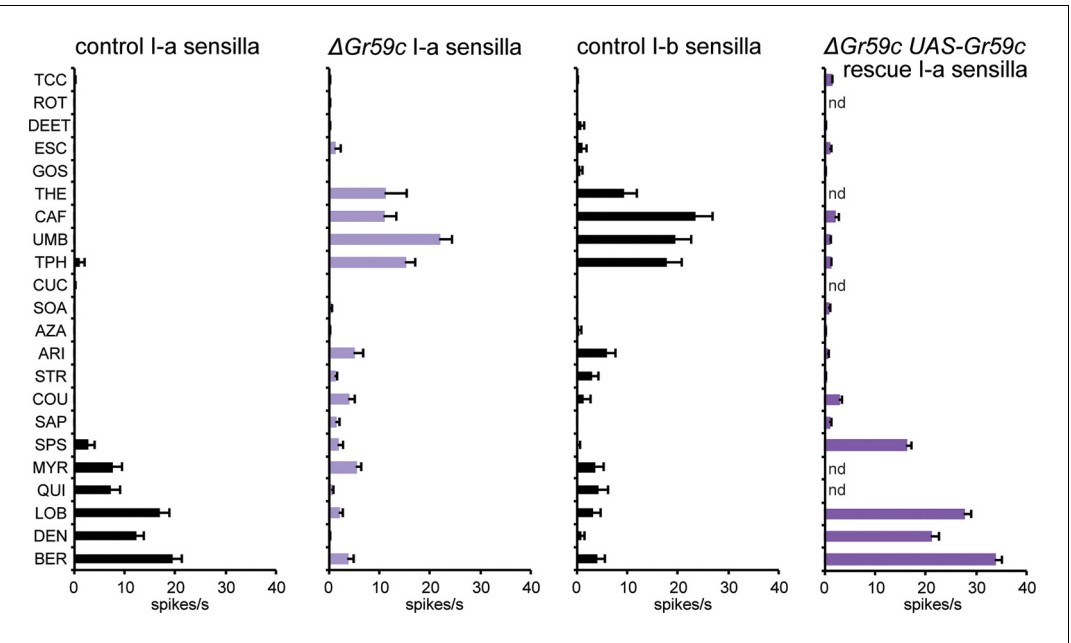

**Figure 7.** Electrophysiological response profiles of *w⁻ Canton-S (wCS)* control I-a sensilla, *ΔGr59c* mutant I-a sensilla, *wCS* control I-b sensilla, and *ΔGr59c* I-a sensilla that had been rescued with a *UAS-Gr59c* construct driven by *Gr66a-GAL4.* Rescued *ΔGr59c* flies were tested with a reduced panel of 16 compounds; the other genotypes were tested with the full panel of 21 compounds.

The following figure supplements are available for figure 7:

**Figure supplement 1.** The *ΔGr59c* mutation was generated through FLP-FRT-mediated recombination between *piggybac* transposon lines *f03881* and *f04393* (*Parks et al., 2004*).

**Figure supplement 2.** Electrophysiological response profiles of *wCS* control S-a sensilla and *ΔGr59c* S-a sensilla.

In the absence of a rescue construct, all flies showed the △Gr59c phenotype (*Figure 9a*), as expected. In the presence of a *UAS-Gr59c* rescue construct with no *GAL80ᵗˢ* construct to repress it, all flies showed a wild-type I-a phenotype, as expected of a rescued *Gr59c* mutant (*Figure 9b*). In the presence of both a *UAS-Gr59c* rescue construct and a *GAL80ᵗˢ* construct, the phenotypes of the sensilla fell into two categories depending on the temperature regime (*Figure 9c*). When the temperature was not increased at all and repression of *UAS-Gr59c* was continuous, a △Gr59c phenotype was observed, as expected, demonstrating that *GAL80ᵗˢ* successfully suppressed *UAS-Gr59c* expression at low temperature (blue bar in *Figure 9c*). However, if the temperature was increased either during pupal development, at 1d after eclosion, or at 5d after eclosion, a wild-type I-a phenotype was restored in every case. Thus, rescue by the *UAS-Gr59c* transgene could occur even after eclosion, in a mature fly. These results argue against a developmental requirement for *Gr59c*. Rather, they are consistent with a physiological role for Gr59c in determining the response profile of the sensillum.

What might be the nature of such a physiological role? Expression of Gr59c increased response to LOB, DEN, and BER in all sensilla tested, and suppressed many other responses (*Figure 6*); deletion of *Gr59c* decreased response to LOB, DEN, and BER and allowed an increase of other responses. One model consistent with these results is that Gr59c binds to one or more co-factors, perhaps co-receptors, to form a complex that responds to LOB, DEN, and BER. The binding to Gr59c would prevent these co-receptors from binding other Grs to form a complex that responds to other tastants. Thus Gr59c might compete strongly with other Grs for binding to essential co-receptors or other co-factors.

This model of Gr59c as a strong competitor might explain why three Grs expressed in I-a (Gr28b.a, Gr28a, and Gr36a) conferred few if any responses. To test this model, we expressed these

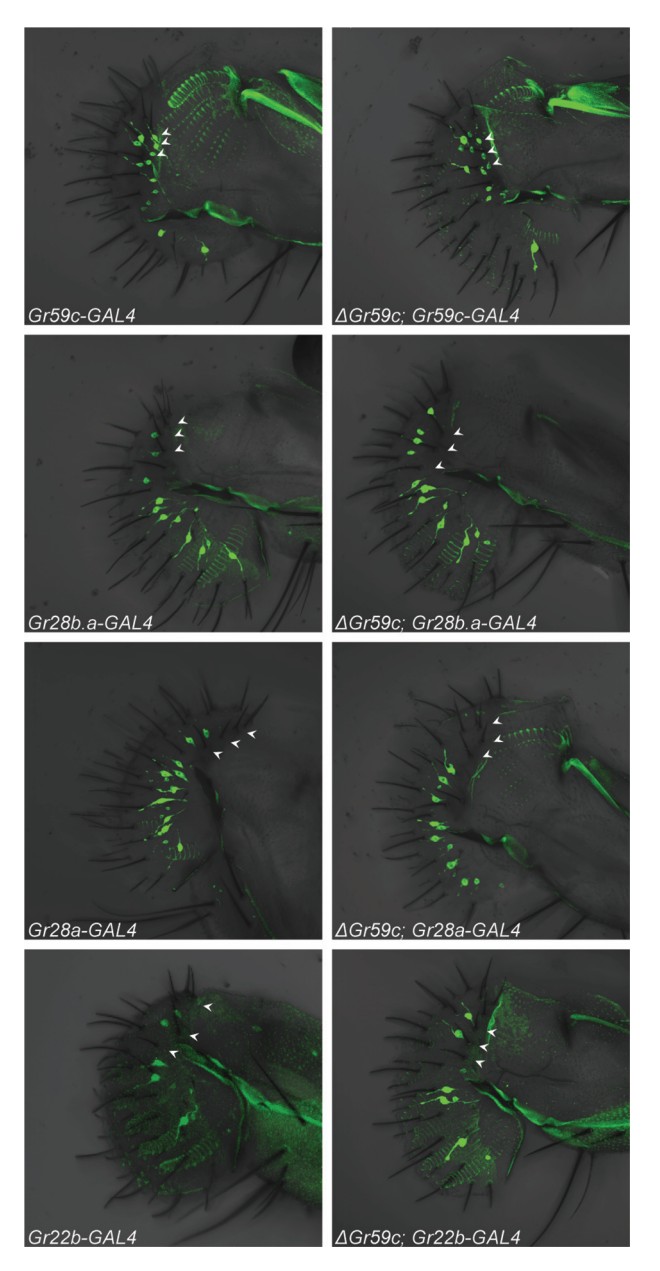

**Figure 8.** Fluorescent confocal microscopy of whole-mount labella reveals that the ΔGr59c mutation does not cause loss of *Gr59c-GAL4* expression in I-a sensilla (top two panels). The ΔGr59c mutation does not cause gain of *Gr28b.a-*, *Gr28a-*, or *Gr22b-GAL4* expression in I-a sensilla (bottom six panels). (n ≥ 5 flies per genotype). White arrowheads indicate the positions of representative I-a neurons. Cell bodies of neurons that innervate I-a sensilla can be seen only in the top two panels, *i.e.* in *Gr59c-GAL4* and ΔGr59c; *Gr59c-GAL4*, as indicated by white arrowheads. Full genotypes tested: *Sp/CyO; Gr-GAL4/UAS-GFP* and ΔGr59c; *Gr-GAL4/UAS-GFP*.

receptors in △*Gr59c* I-a sensilla, where the putative strong competitor is absent, to determine if they conferred stronger responses.

## Grs confer responses to △*Gr59c* I-a sensilla that are not conferred to wild-type I-a sensilla

Expression of Gr28a in △*Gr59c* I-a sensilla conferred response to eight tastants (*Figure 10*). By contrast, expression of Gr28a in wild type I-a sensilla increased responses to none of these tastants

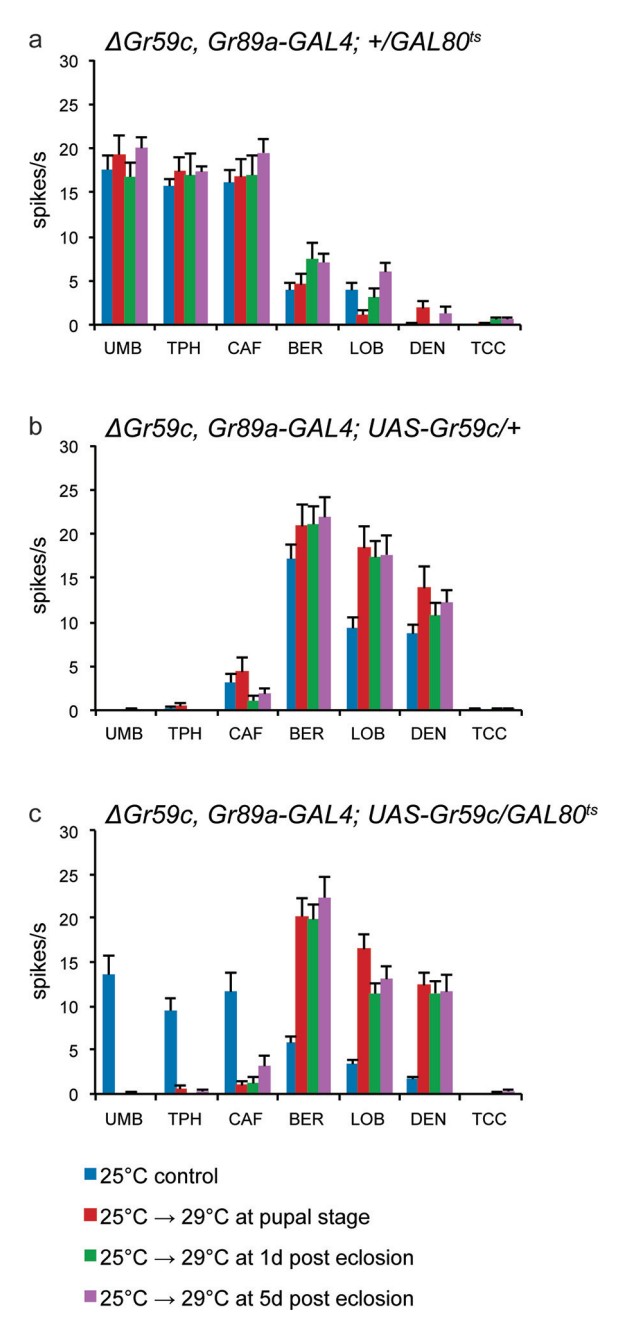

**Figure 9.** *UAS-Gr59c* expression in adult flies is sufficient to restore wild-type responses to *ΔGr59c* mutant I-a sensilla. All genotypes have identical 2nd chromosomes: *ΔGr59c, Gr89a-GAL4.* Flies were subjected to the four indicated temperature regimes. (a) *GAL80ts* parental control flies without *UAS-Gr59c* display mutant, elevated responses to UMB, TPH, and CAF in all regimes. (b) *UAS-Gr59c* parental control flies without *GAL80ts* in all cases display wild-type responses to BER, LOB, and DEN. (c) Experimental flies (*UAS-Gr59c/GAL80ts*) display mutant responses when kept continuously at *GAL80ts* permissive temperature (25°C), indicating that *GAL80ts* is suppressing *UAS-Gr59c* expression. However, when *GAL80ts* is inactivated at any time by shifting flies to higher temperature (29°C), thus activating *UAS-Gr59c*, flies display wild-type responses.

(*Figure 3a*). Expression of Gr36a in △*Gr59c* I-a sensilla conferred responses to 6 tastants, while suppressing responses to two. By contrast, expression of Gr36a conferred no changes in response profiles of wild type I-a sensilla (*Figure 3a*), or S-a or I-b sensilla (*Figure 4a,b*). Expression of Gr28b.a

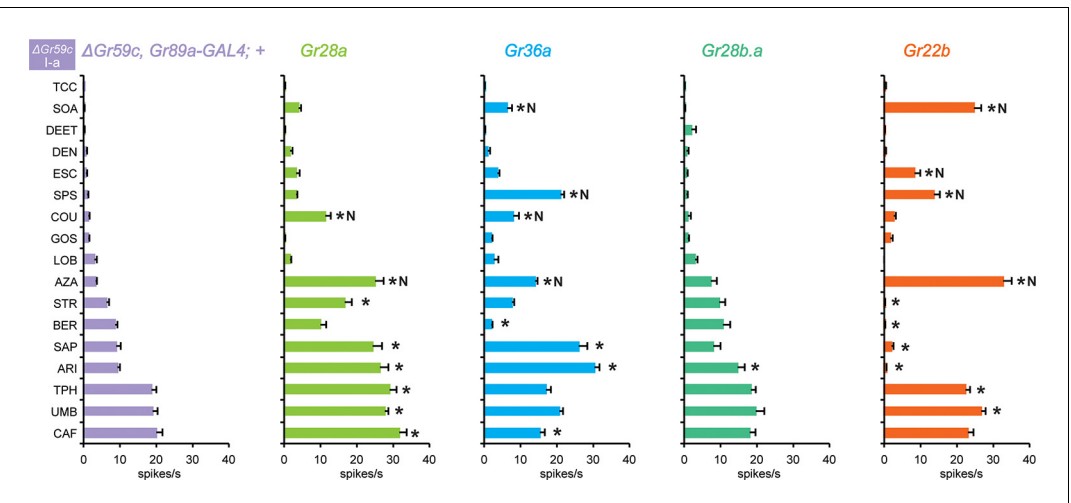

**Figure 10.** Electrophysiological responses in sensilla ectopically expressing four individual Grs, in *ΔGr59c* I-a bitter neurons. Novel responses are indicated by 'N'. The experimental genotypes are: *ΔGr59c, Gr89a-GAL4; UAS-GrX*, except that in the case of Gr22b, the experimental genotype is *ΔGr59c; UAS-Gr22b/Gr66a-GAL4*. Asterisks indicate significant changes relative to the *ΔGr59c, Gr89a-GAL4; +* parental control. $p \leq 0.0001$, except that $p \leq 0.0005$ for Gr36a/SOA and Gr22b/STR; $p \leq 0.01$ for Gr36a/CAF, Gr22b/TPH; $p \leq 0.05$ for Gr28b.a/ARI. $n \geq 24$).

conferred an increased response to ARI, which is consistent with the increased response to ARI found in I-b sensilla (*Figure 4a,b*; MYR was not tested).

Gr22b had conferred strong, novel responses to CUC, SOA and AZA in I-a (*Figure 3a*), and we wanted to determine if in the absence of Gr59c these responses were still conferred. We tested two of these responses, SOA and AZA, and both were conserved (*Figure 10*). Responses were also conferred to ESC and SPS, and several responses were suppressed.

In summary, removal of *Gr59c* from I-a sensilla allowed Grs to confer novel responses that were not observed in the presence of *Gr59c*. These results provide further support for the hypothesis that the function of bitter Grs depends on the molecular context. The results are also consistent with the notion that Grs may interact with each other or compete for common co-factors.

## Discussion

### In vivo expression analysis of Grs in a natural context

We have carried out a systematic functional analysis of eight *Gr* genes, most of which have not been examined functionally before. While most previous analysis of bitter *Gr* genes has been accomplished via loss-of-function analysis, we have examined all eight *Gr* genes through in vivo expression analysis. We expressed them in bitter-sensitive neurons that either do or do not express them in wild type, *i.e.* overexpression or ectopic expression, respectively. These bitter-sensitive neurons also express different patterns of other bitter Grs. We have complemented some of this analysis via loss-of-function analysis as well.

All of our studies have been carried out in vivo, in a natural laboratory: in taste neurons in gustatory sensilla of the labellum, which have evolved over the course of hundreds of millions of years to sample, detect, and signal taste information. The taste stimuli have been delivered by direct contact in a manner that simulates natural sampling of tastants, and the responses have been measured by recording action potentials, the signals transmitted to the brain.

All expressed Grs had functional effects in at least one neuronal type. This uniformity is noteworthy in part because of the diversity of the tested Grs. Although most of them are expressed in the labellum of wild type, as determined by *Gr-GAL4* analysis, some are expressed in the legs, the pharynx, the larval taste system, and even the antenna, where taste receptor function is enigmatic (*Weiss et al., 2011*; *Kwon et al., 2011*; *Kwon et al., 2014*; *Scott et al., 2001*; *Fishilevich and*

*Vosshall, 2005*; *Menuz et al., 2014*; *Fujii et al., 2015*). Moreover, the tested receptors are dispersed widely not only across tissues and developmental time, but also across the phylogenetic tree of Grs (*Robertson et al., 2003*).

The effects of nearly all Grs, moreover, were easily distinguishable. There was one notable exception, however: the profiles of Gr2a and Gr10a were very similar in each of three sensillum types examined (*Figures 3*, *5b,d*). *Gr2a* and *Gr10a* are not recently duplicated genes, but rather are on distant branches of the *Gr* phylogenetic tree (*Robertson et al., 2003*). We have previously found that similar odor response spectra are conferred by the *D. melanogaster* and *D. pseudoobscura* orthologs of Or71a, which contain only 59% amino acid sequence identity (*Ray et al., 2008*). Gr2a and Gr10a provide an even more extreme case: they exhibit only 32% identity.

## Grs confer different effects in different neurons

A key finding of this study is that expression of individual Grs produced different effects in different bitter taste neurons (*Figure 11*, *Figure 11—figure supplement 1*). This finding supports an emerging theme in the field of taste: that bitter taste neurons are heterogeneous (*Weiss et al., 2011*; *Meunier et al., 2003*; *French et al., 2015*; *Glendinning et al., 2006*; *Glendinning et al., 2002*). Historically, a long-held concept in the field has been that all bitter taste neurons are identical, and that they function identically to warn the animal indiscriminately of toxic compounds (*Freeman and Dahanukar, 2015*). An emerging concept is that bitter taste neurons are distinguishable, both in terms of response profile and receptor expression (*Weiss et al., 2011*; *Ling et al., 2014*; *Meunier et al., 2003*). The differing effects of expressing an individual Gr in varying neuronal contexts adds another line of evidence to support the view that different bitter taste neurons are functionally distinct, thereby endowing the system with richer coding capacity.

Expression of Grs produced three kinds of effects: a novel response to a particular tastant, not observed in either parental control; an increased response to a tastant, greater than that observed in controls; a decreased response to a tastant, *i.e.* suppression. All three effects were observed broadly, and most receptors exhibited all three kinds of effects (*Figure 11*, *Figure 11—figure supplement 1*). However, there was specificity to the effects, in that different receptors conferred varying effects in different neurons or in response to different tastants. The degree of diversity was striking, in the case of both sensilla and receptors. For example, in the S-a sensillum, expression of Gr22b increased some responses and decreased others; Gr58c only decreased responses; Gr36a had no effects on any responses. In the case of the Gr10a receptor, which is expressed in the antenna as well as a larval taste organ, expression in S-a sensilla increased some responses and

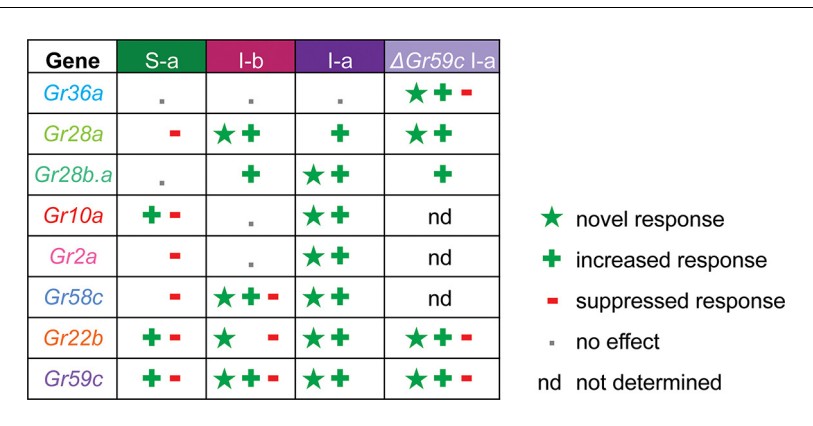

**Figure 11.** Gr ectopic expression produces receptor-specific and neuron-specific effects on the response profiles. Effects are delineated for each tastant in *Figure 11—figure supplement 1*.

The following figure supplement is available for figure 11:

**Figure supplement 1.** Summary of effects of *UAS-Gr* expression on responses to tastants in different sensillum types, based on comparison with tested control lines.

decreased others; expression in I-a produced increased responses and novel responses but no decreased responses; expression in I-b had no effects.

## Models of direct or indirect interaction between bitter Grs

According to the simplest model of Gr function, one might have expected that expression of a Gr would simply add to the response profile of the neuron. If the expressed Gr responded to a tastant that the endogenous Grs in the native neuron do not respond to, then a novel response would be added to the profile. If the expressed Gr responded to a tastant that the endogenous Grs respond to, then the responses to that tastant would then increase.

The effects observed in this study are difficult to reconcile with this simple model of Gr function. A more complex model is required to explain, for example, the findings that: i) expression of some Grs decreased endogenous responses; ii) deletion of *Gr59c* increased the response to certain tastants; iii) expression of some Grs in neurons in which they are endogenously expressed caused novel responses; iv) the effects of expression of a Gr differed among neuron types (*Figures 11*, *12*, *Figure 11—figure supplement 1*). Below we consider mechanisms that could explain these findings.

### Suppression of endogenous responses

Expression of nearly all Grs decreased at least some responses, in at least one kind of neuron (*Figure 11*). Grs suppressed responses either when expressed ectopically or when overexpressed in a cell. We note that Ors can confer inhibitory as well as excitatory responses to olfactory receptor neurons (*Hallem et al., 2004*). *A priori,* one simple model for how a Gr could decrease the response of a neuron to a particular tastant is by binding it and inducing a current or a signal that counteracts the activity of another, endogenous Gr that binds the same tastant and independently excites the neuron. However, Grs decreased responses in a neuron-specific manner. For example, expression of Gr22b decreased the response to DEN in S-a but increased it in I-a. If the function of Gr22b were uniquely to bind DEN and produce an inhibitory response, it is difficult to explain how it would increase the response to DEN in I-a.

A second model for how some bitter Grs can suppress neuronal response is by competing with other Grs for a necessary signaling factor. For example, Grs might compete for binding to a common co-factor. In the olfactory system, most ORNs express one of ~60 Ors that are believed to bind odorants, along with a common co-receptor, Orco, which is required for function of the Or (*Larsson et al., 2004*; *Vosshall and Stocker, 2007*). Moreover, an RNA-Seq analysis was consistent with a 1:1 stoichiometry of Ors and Orco in the antenna (*Menuz et al., 2014*). It is conceivable that some bitter Grs compete within a taste cell for binding to a common co-factor. If so, some Grs might be more effective competitors than others, which could explain in part why expression of some Grs conferred many, strong suppressive responses, while others conferred few if any. This model does not specify the number or nature of co-factors, does not require a co-factor to be expressed in all bitter neuron types, and does not specify whether the co-factor acts in membrane-targeting, ligand-binding, or other functions.

A third model for suppression, related to the second and compatible with it, is that a bitter Gr inhibits another Gr directly by binding to it and inactivating it (*Figure 12, i*). Some bitter Grs may bind a tastant, conferring an excitatory response to a taste neuron, and also bind to and inactivate another Gr. Thus overexpression of a Gr in a neuron might increase the response of the neuron to the tastant that the Gr binds, but decrease the response mediated by another Gr to another tastant. Perhaps Gr59c, for example, confers a response to BER, LOB, and DEN directly, but inhibits a Gr that confers a response to TPH, UMB, and CAF, consistent with earlier results (*Weiss et al., 2011*). This model would explain the paradoxical finding (*Figure 12, ii*) that loss of a receptor, Gr59c, led to an increase in response to TPH, UMB, and CAF.

### Novel and increased responses

The simplest model for how expression of a Gr might confer a novel or increased response to a tastant is by binding to it and transducing the signal so as to excite the neuron. Such excitation could confer a novel response or an increased response, depending on whether the neuron exhibited an endogenous response to the tastant.

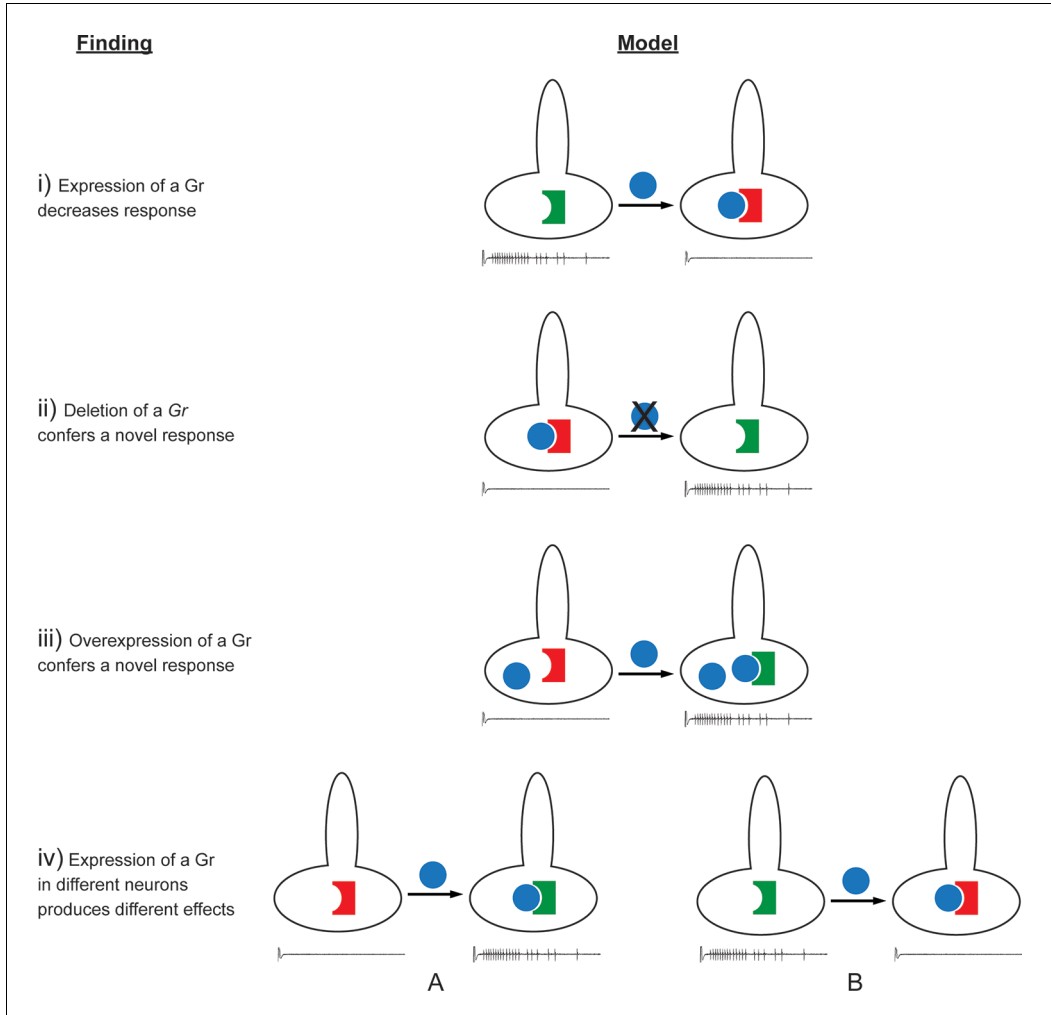

**Figure 12.** Findings and models. Four findings of this study are indicated, along with one possible model to explain each. **i)** Expression of a Gr, indicated by the blue sphere, decreases the response to a tastant, represented by the traces below. One possible model is that the expressed Gr (blue sphere) interacts with another, active Gr (green) and inhibits it (represented by its conversion from a green, active Gr to a red, inactive Gr). **ii)** Deletion of *Gr59c* (blue sphere with X) leads to an increased response. One possible model is that in wild type, the Gr inhibits an endogenous Gr (red). Removal of the Gr allows the inactive Gr (red) to become active (green). **iii)** Overexpression of a Gr, in a neuron that contains the Gr endogenously, induces a response that is not observed in wild type. One model is that above a certain concentration threshold of the Gr, it is able to bind and convert another Gr from an inactive (red) to an active (green) form. **iv)** Expression of a Gr in two different neurons, 'A' and 'B,' produces different results. The neuron at left shows an increase in response to a tastant whereas the neuron at the right shows a decrease. One model is that in the A neuron, the expressed Gr (blue) binds to an inactive Gr (red) specific to neuron A, and activates it (green). By contrast, in the B neuron, the expressed Gr (blue) binds to a different Gr (green) specific to neuron B, and inactivates it (red).

This simple model does not easily explain how expression of some Grs in neurons that express them endogenously, *i.e.* 'overexpression,' led to novel responses (iii). One model, related to the third model of suppression, again entails the binding of one Gr to another Gr. In this case, however, binding would result not in inhibition but in activation (*Figure 12, iii*). The binding of two Grs could result in an activated Gr—perhaps a heteromer of two or more subunits—that has a function not observed in the native neuron. According to this model, the response profile of the neuron would be sensitive to the relative levels of Grs; only after the level of a Gr exceeded a certain threshold by virtue of overexpression would levels of the heteromer become detectable in functional tests.

The concept of subunit interactions could also explain the finding that expression of individual Grs conferred different effects on different neurons (*Figure 12, iv*). Since each different neuron type contains a different roster of Grs, the interplay of subunits could easily differ in each type, leading to a different pattern of inhibition and activation of responses.

The possibility of interactions among bitter Grs, and in particular the formation of Gr heteromers, is consistent with the results of several previous studies. First, expression of single bitter Grs in sugar-sensitive neurons or heterologous expression systems has not yielded bitter responses, consistent with the possibility that a combination of bitter Grs is required (*Liman et al., 2014*). Second, the responses to some bitter compounds were reduced by mutation of any of multiple Grs, consistent with the notion that these Grs combine to form a functional multimer (*Lee et al., 2009*; *2010*; *2012*; *Moon et al., 2006*; *2009*). In the antenna, two Grs, Gr21a and Gr63a, are coexpressed and together confer response to carbon dioxide (*Jones et al., 2007*; *Kwon et al., 2007*). There is evidence for subunit interactions among sugar receptors (*Fujii et al., 2015*), although elegant experiments have shown that responses conferred by expression of individual sugar receptor genes in an antennal neuron agree well with the responses lost by mutations of the same individual *Gr* genes (*Freeman et al., 2014*).

## Implications for the evolution and modulation of toxin detection

The diverse effects we have observed from expressing Grs may have profound evolutionary implications. A major challenge for *Drosophila* and other organisms is to respond to changes in the chemical environment. For example, as new plants move into the habitat of a fly, or as a fly moves into a new habitat, there are novel opportunities for nutrition but also novel dangers of toxicity. How can the fly adapt so as to exploit the opportunities but avoid the dangers? The ability of its taste system to encode and interpret new chemical signals is crucial. A classic means of detecting new signals is via changes in the amino acid sequence of taste receptors (*Ueno et al., 2001*). A change in the binding site of a bitter receptor, for example, may allow the detection of a new toxin.

Our results suggest a rich, alternative means of generating variation in chemosensory signaling. We have found that changes in the expression of a taste receptor can generate a wide variety of effects: increased responses, decreased responses, and novel responses. Accordingly, simple alterations in regulatory sequences that govern the cellular specificity of expression, or the level of expression, could change the response specificities of taste neurons in dramatic and beneficial ways. Thus, the organization of the taste system—featuring the coexpression of many Grs in individual neurons—may provide numerous degrees of freedom for altering the response patterns of the neurons, via either the gain or loss of receptor function. Such alterations could meet a specific environmental challenge, or could in a more general sense expand the potential of the system to encode and perhaps distinguish among bitter compounds of differing toxicity or significance.

A network of coexpressed Grs, sensitive to relative levels of expression, could also provide a means of modulating taste sensitivity in response to short-term changes in either the environment or the internal state of the fly. We found that alterations in the expression of *Gr59c* in the mature fly induced changes in taste sensitivity (*Figure 9*). Thus, short-term changes in *Gr* transcription or post-transcriptional processing could in principle be a mechanism of altering taste response to meet the short-term needs of a hungry or satiated fly (*Prestia et al., 2011*; *Zhou et al., 2014*).

To investigate whether alterations in Gr expression may provide a mechanism for the evolution or modulation of taste responses, it will be of interest to examine the behavioral consequences of such alterations. For example, does the expression of Gr22b in I-a, I-b, or S-a sensilla, which produces increased or novel electrophysiological responses to SOA, increase the fly's behavioral sensitivity or degree of aversion to SOA? Does expression of Gr58c in I-b and S-a, which suppresses electrophysiological responses to UMB, decrease the fly's behavioral response to UMB? Increased aversion to a bitter tastant in a food source could be adaptive in environments that contain a particularly rich selection of more beneficial food sources. Decreased aversion to a bitter compound in a food source might be adaptive following the starvation of a fly with few alternatives.

In a more general sense, the functional organization of the taste system differs from that of the olfactory system in that the 60 *Gr* genes are more highly coexpressed in a smaller number of neuronal types than the 60 *Or* genes. This pattern of organization may in principle entail some loss of discriminatory power, but it could provide a degree of evolutionary and regulatory flexibility that may serve the needs of the animal well.

Finally, we note that the present analysis of eight *Gr* genes lays a genetic foundation for a detailed biochemical analysis. It will be of interest, for example, to test the possibility of association between tagged Grs in vivo. The results also provide a framework and a focus for further testing of specific hypotheses about the functions of individual *Gr* genes, via the construction of a series of genotypes with CRISPR/Cas technology (*Bassett et al., 2013*; *Gratz et al., 2013*).

## Materials and methods

### *Drosophila* stocks

Flies were grown on standard cornmeal-agar medium. Flies used for physiological recordings and imaging were grown at 25°C in a humidity-controlled incubator. Only females, aged 6–8 days, were used for electrophysiological recording. Male and female flies, aged 7–14 days, were used for imaging. All transgenic genotypes tested were homozygous, except where indicated.

### Transgenic constructs

*Gr-GAL4* drivers (*Gr89a-GAL4, Gr66a-GAL4, Gr59c-GAL4, Gr28a-GAL4, Gr28b.aGAL4, Gr22b-GAL4*) were as described in *Weiss et al. (2011)*; *Gr28a-GAL4* was provided by H. Amrein. *UAS-Gr* lines were generated by amplification of *Gr* coding regions from Canton-S cDNA prepared from proboscis, legs, and larvae. *UAS-Gr2a, -Gr22b, -Gr28a, -Gr36a, -Gr59c* were cloned into the pUAST expression vector and inserted into the genome using random transposon integration. *UAS-Gr10a, -Gr28b.a*, and *-Gr58c* were cloned into the pBI-UASCG expression vector and inserted into the genome with *phiC31* site-specific integration into the following strains: 9725 (*Gr10a*), 9744 (*Gr28b.a*), and attP2 (*Gr58c*) (*Groth et al., 2004*; *Pfeiffer et al., 2010*). Since *UAS-Gr2a* and *UAS-Gr10a* gave very similar results, we verified with PCR that the lines containing these constructs in fact contained the expected transgenes.

### *Gr59c* deletion

The Δ*Gr59c* line was generated using FLP-FRT mediated recombination (*Parks et al., 2004*) between two FRT-bearing *piggybac* transposons from the Exelixis collection (*pBacf04393* and *pBacf03881*) flanking a ~17 kb genomic region encompassing the *Gr59c* locus (see *Figure 7—figure supplement 1*). The deletion was confirmed through PCR and is homozygous viable. The Δ*Gr59c* line was backcrossed into *w⁻ Canton-S* flies (*wCS*) for 7 generations. We note that a mutant phenotype different from that shown in *Figure 7* was observed in the original Exelixis $w^{1118}$ isogenic background before the outcrossing to *wCS*; the phenotype shown in *Figure 7* was also observed in all other genetic backgrounds and contexts tested (n>10).

### Taste compounds

Tastants were obtained at the highest available purity from Sigma-Aldrich. All tastants were dissolved in 30 mM tricholine citrate (TCC), an electrolyte that inhibits the water neuron (*Wieczorek and Wolff, 1989*). Tastant solution aliquots were stored at −20°C long-term and kept at 4°C while in use, for no more than a week. See *Table 1* for concentrations used.

### Electrophysiology

Single-sensillum recordings were performed as described in Delventhal et al. (*2014*). To quantify responses, the number of action potentials (spikes) was counted over a 500 ms period, starting 200 ms after contact. A high salt stimulus (400 mM NaCl), which activates the bitter neuron (*Hiroi et al., 2003*; *Meunier et al., 2003*) was used as a positive control at the beginning and end of the recording session for each sensillum to ensure that the bitter GRN was functional. All recordings from sensilla that displayed an average NaCl response of less than 10 spikes/s at the beginning or end of a recording session were discarded. No more than eight tastants were tested on an individual sensillum of a given fly, with 2–3 minutes between presentations.

### Confocal imaging

Whole labella were dissected and their GFP fluorescence was imaged using a Zeiss LSM510 confocal microscope under 40X magnification. Images were processed using ImageJ.

## Statistical analysis

All error bars represent S.E.M. Experimental genotype response profiles were compared to the *GrX-GAL4* (and *UAS-GrX*, when available) parental control response profiles with a two-way ANOVA and a Bonferroni multiple comparisons correction, using Prism software. A one-way ANOVA was performed within the parental control genotype to determine which responses were significantly different from the TCC diluent control. Responses that were determined to be significantly greater than TCC within the control genotype were then designated as 'increased' when elevated in experimental genotypes; responses that did not differ from TCC in the control response profile were designated as 'novel' when found to be elevated in experimental genotype profiles.

## Acknowledgements

We thank members of the Carlson Laboratory and Fred Marion-Poll for helpful discussion and comments on the ms. We thank Linnea Weiss for generating some of the *UAS-Gr* lines.

## Additional information

### Funding

| Funder | Grant reference number | Author |
| --- | --- | --- |
| National Institute on Deafness and Other Communication Disorders | 5F31DC012985 | Rebecca Delventhal |
| National Institute on Deafness and Other Communication Disorders | | John R Carlson |

The funders had no role in study design, data collection and interpretation, or the decision to submit the work for publication.

### Author contributions

RD, Conception and design; Acquisition of data; Analysis and interpretation of data; Drafting or revising the article; Contributed unpublished essential data or reagents; JRC, Conception and design; Analysis and interpretation of data; Drafting or revising the article

## Additional files

### Supplementary files

• Supplementary file 1. (A) Responses of I-a sensilla recorded from flies of the indicated genotypes. (a) Mean spikes/s. (b) S.E.M. (c) n, where n represents the number of traces analyzed. (B) Responses of I-b sensilla recorded from flies of the indicated genotypes. (a) Mean spikes/s. (b) S.E.M. (c) n, where n represents the number of traces analyzed. (C) Responses of S-a sensilla recorded from flies of the indicated genotypes. (a) Mean spikes/s. (b) S.E.M. (c) n, where n represents the number of traces analyzed. (D) Responses of ΔGr59c I-a sensilla recorded from flies of the indicated genotypes. (a) Mean spikes/s. (b) S.E.M. (c) n, where n represents the number of traces analyzed.

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
