## [Decision Letter]

Thank you for submitting your work entitled "Bitter taste receptors confer diverse functions to neurons" for consideration by *eLife*. Your article has been reviewed by two peer reviewers, and the evaluation has been overseen by a Reviewing Editor and a Senior Editor.

The reviewers (Yehuda Ben-Shahar and one other) have discussed the reviews with one another and the Reviewing editor has drafted this decision to help you prepare a revised submission.

Summary:

This paper represents a comprehensive analysis of the role of "bitter" receptor choice in determining the ligand specificity of labellar gustatory receptor neurons in the fruit fly. Overall, this is an important contribution to our understanding of the insect taste system. It is the first report that sheds light on how bitter taste ligand specificity might be determined. The novel and unintuitive results suggest an unappreciated complexity to Gr function that will require investigation.

Essential revisions:

The paper is missing the behavioral context, which could clarify the possible ecological significance of some of the more puzzling results it reports. The reviewers noted that it would be of interest to assess whether alterations of physiological response profiles as a consequence of ectopic receptor expression have in fact behavioral consequences. While such measurements are felt to be outside the scope of the present study, they do deserve some discussion. This would also help build up the potential evolutionary implications of this study.

---

## [Author Response]

*Essential revisions:*

*The paper is missing the behavioral context, which could clarify the possible ecological significance of some of the more puzzling results it reports. The reviewers noted that it would be of interest to assess whether alterations of physiological response profiles as a consequence of ectopic receptor expression have in fact behavioral consequences. While such measurements are felt to be outside the scope of the present study, they do deserve some discussion. This would also help build up the potential evolutionary implications of this study.*

We are grateful for the helpful suggestion that we discuss the issue of whether the alterations of physiological response profiles have behavioral consequences. We agree that this point is highly germane to the potential evolutionary implications of the study.

Accordingly, we have added a paragraph to the Discussion that directly bears on this issue:

“To investigate whether alterations in Gr expression may provide a mechanism for the evolution or modulation of taste responses, it will be important to examine the behavioral consequences of such alterations. For example, does the expression of Gr22b in I-a, I-b, or S-a sensilla, which produces increased or novel electrophysiological responses to SOA, increase the fly’s behavioral sensitivity or degree of aversion to SOA? Does expression of Gr58c in I-b and S-a, which suppresses electrophysiological responses to UMB, decrease the fly’s behavioral response to UMB? Increased aversion to a bitter tastant in a food source could be adaptive in environments that contain a particularly rich selection of more beneficial food sources. Decreased aversion to a bitter compound in a food source might be adaptive following the starvation of a fly with few alternatives.”